# Experimental Study of Filtration Materials Used in the Car Air Intake

**DOI:** 10.3390/ma13163498

**Published:** 2020-08-07

**Authors:** Tadeusz Dziubak, Sebastian Dominik Dziubak

**Affiliations:** Faculty of Mechanical Engineering, Military University of Technology, gen. Sylwestra Kaliskiego Street 2, 00-908 Warsaw, Poland; sebastian.dziubak@wat.edu.pl

**Keywords:** filter medium, engine, particle size, nanofibers, separation efficiency, filtration performance, pressure drop, dust mass loading

## Abstract

Traditional cellulose filter media used for air filtration in vehicle engines are characterized by 99.9% filtration efficiency and accuracy above 2–5 µm. The highest engine component wear is caused by dust grains above 1 µm. Filter media with nanofiber additions provide greater filtration efficiency of dust grains below 5 µm. Filter material selection for vehicle engine air filter is a problem because their manufacturers mainly provide only the structure parameters: pore size, air permeability, and thickness. There is no information about material filtration properties using polydisperse test dust. The manuscript presents methodology and experimental test results of five samples A, B, C, D and E, filter materials differing in their chemical composition and structure parameters. In the first stage, efficiency characteristics *φ_w_*, filtration accuracy *d_zmax_* and the flow resistance Δ*p_w_* depending on the dust absorption coefficient *k_m_* of three filter cartridges of each material, A, B, C, D and E, were determined. Then, from each material characteristics of one piece was selected in order to compare their initial and initial period efficiencies as well as changes in the flow resistance depending on the dust absorption coefficient *k_m_*. Obtained results showed that the filter materials differ significantly in efficiency and accuracy values in the initial filtration period. Initial period duration is also different, i.e., filtration efficiency increasing time to a certain value, which for materials with a nanofiber layer is much shorter, which minimizes engine component wear. For materials with nanofibers, flow resistance increase intensity is greater, which results from surface filtration. Filtration efficiency of each filter material sample A, B, C, D and E was assessed with the filtration quality coefficient including the efficiency and flow resistance. In the available literature, the problem of increasing filtration efficiency in the initial period is known, but there are no results for specific filter materials. Research shows that filter material characteristics are closely related. Each increase in efficiency and accuracy of intake air filtration reduces engine components wear, but it is related to flow resistance increase in the engine intake system, which reduces its power, and increases need for more frequent filter servicing.

## 1. Introduction

Road surfaces are a place of particle deposition from many different sources. First and foremost, it is mineral dust blown by the wind from the soil surrounding the road, from the sand and desert areas that cover about one third of the earth’s surface [1]. Those include dust generated during field works, road works or construction works. The particles may include salt used for de-icing, plant matter, pollen, animal hair, and other biological matter from the surrounding areas [2]. Apart from the natural processes, significant amount of dust particles is formed as a result of human activities, including agriculture, industrial production and mining [3]. Contamination from anthropogenic sources include the particles released to the atmosphere as a result of processes between brake and coupling components [4,5,6] and tire contact with the road [7,8,9]. Those include particles from the engine’s exhaust gases including soot, and engine components’ wear products.

The main component of particles settled on the road is mineral dust, containing solids formed as a result of material processing, including crushing, milling, hammering, reloading, detonation and destruction of organic and inorganic matter, including rock, ore and metal. The friction created by the tire of a moving vehicle on the unpaved and dusty paved road or by the wind, lift the particles from the road surface and moves them around as the road dust [10,11]. The main component of the road dust is silica (SiO_2_)—between 60–95%. The other components (4–19%) include different metal oxides: Al_2_O_3_, F_2_O_3_, CaO, MgO and organic matter [12,13].

The air from the atmosphere is the basic working medium of all internal combustion engines used in motor vehicles. The combustion of 1 kg of petrol requires at least 14.5 kg of air. During operation in nominal conditions, the internal combustion engine in passenger car requires 150–400 m^3^/h of air and, in trucks, 900–1400 m^3^/h. The Leopard 2 tank engine requires over 6000 m^3^/h of air.

The internal combustion engines suck large amount of particles with the air. These are impurities with grain sizes not exceeding dz = 80–100 µm. A characteristic feature of polluted air is the concentration of dust in the air *s* (g/m^3^). It has different values depending on the conditions and the ground on which the vehicle is traveling. The lowest dust concentration in the air (s = 2–20 mg/m^3^) occurs on paved streets and roads, and the highest (up to 10 g/m^3^) occurs when tracked vehicles are driven on training grounds with dry ground [12].

The most dangerous air contaminant for machinery is the mineral dust with high silica (SiO_2_) content and a hardness of 7 on the Mohs scale. Silica is the main cause of high engine component wear including piston rings, cylinder sleeves, crankshaft main bearing journals and sleeves, valves and valve guides [14]. A minimum service life of those components determines engine life.

In turbine engines, there are no reciprocating components or slide bearings. High air flow rates 150–250 m/s, high exhaust gas rates (over 300 m/s) and high tangential velocity (200–500 m/s) allow large dust particles to achieve high kinetic energy at the point of contact with the compressor rotor blades, and may cause severe impact. The impact may cause premature erosion wear due to removal of metal microparticles from the component’s surfaces, damage the surface structure and change its shape. As a result, engine efficiency and durability may be affected.

Smaller dust particles accumulate on the compressor body walls, particularly in the corners and on the guide vanes. Some dust particles with a significantly lower melting point (Sodium Chloride NaCl-1074K, Albite NaAlSi_3_O_8_-1388K) compared to the average combustion temperature (1600 K) are melted in the combustion chamber; however, their inertia is sufficient to let them accumulate on the duct or rotor surface. The rate at which the dust layer accumulates, chemical composition of vitreous material (lime-magnesium-aluminosilicate deposits on the turbine guide vanes), and the erosion of compressor blades can reduce the mass flow rate through the engine and affect its performance: reduced power and increased fuel consumption, and as a result affect the flight safety [15].

Tests showed that the engine durability depends on the purity of inlet air [16,17,18]. The wear of engine components is caused by 1–40-µm dust particles; however, the most damage is done by 5–20-µm particles [19,20,21,22,23,24].

Excessive wear of piston rings and cylinder sleeves results in reduced compression pressure and engine power, and may increase the blow-by to the crankcase [25,26,27]. Some impurities (approx. 30%) supplied with the air to the engine cylinders are removed with the exhaust gases, increasing PM emission by the engine [28].

Inlet air filtration systems in modern passenger car engines use single-stage filters with panel filters made of pleated filter paper. Trucks, heavy vehicles and other vehicles operated in high-dust concentration conditions are usually equipped with two-stage filtration systems [29,30,31].

The first stage is an inertial filter, a multi-cyclone or a mono-cyclone with vortex wheel, and the second filtration stage is a cylindrical porous panel filter made of pleated filter paper.

A predominant filter medium used in the inlet air filters in modern combustion engines are filter papers (porous material), characterized by separation efficiency *d_z_* ≥ 5 µm, filtration performance of *φ_w_* = 99.9%, low thickness *g_m_* = 0.4–0.8 mm, and small (*k*_m_ = 200–250 g/m^2^) dust mass loading limited by a permissible pressure drop Δ*p_fdop_* of the air filter [32,33]. Filter paper retains dust particles at the fibers of a porous filter media as a result of different forces and separation mechanisms [34,35].

The inertial filter is characterized by the ability to separate large masses of dust from large air streams with an separation efficiency of 85–95% and not very high filtration performance (over 15–35 µm) without changing its flow resistance and the need for maintenance. Thus, a much smaller mass of dust gets on the paper element, which results in an extended service life of the air filter [18,20,21].

Over time, the dust particles settle deep into the fibrous structure of the filter media, preventing air flow, resulting in pressure drop (increase in flow resistance Δ*p_f_*). It increases the additional costs of energy used to force the air to flow and reduce the filling ratio and engine power.

Cellulose filter media commonly used in inlet air filters in motor vehicles are made of fibers with relatively large diameters, usually over 10 microns. Many researchers show that reducing the fiber diameter in the filter bed increases the separation efficiency. Using 1- instead of 50-μm fibers increases the separation efficiency 2000 times [36].

Separation efficiency mechanisms are not affected by the particle velocity at the inlet but by the relationship between the particle and fiber size and increases with an increase in particle size and a decrease in fiber diameter.

This phenomenon can be implemented by using polymer nanofibers, i.e., fibers with a diameter below 1 µm. A thin layer of nanofibers applied from the inlet side on a standard filter medium (e.g., cellulose) will retain the particles before reaching the inside of the filter medium. The dust retained at the surface of the nanofiber layer can be easily removed (filter cleaning) with an impulse of compressed air in the opposite direction to the air flow during normal operation. If the dust particles are accumulated on the surface of the filter medium, removing them under high pressure will not cause damage to the structure of the filter element.

Compared to the standard fibers, nanofibers show unique and new properties, for example, high surface area and significantly higher strength relative to their mass and higher chemical activity and moisture sorption. The term “nanofibers” is used for fibers manufactured using an “electrospinning” technology [37,38,39,40,41,42,43,44]. Nanofibers with very small diameters, approx. 50–800 nm, are used in the automotive industry. The manufacturers of filter media using nanofibers have developed proprietary technologies, e.g., Ultra-Web^®^ and Fibra-Web^®^ by Donaldson Company Inc. Minneapolis, MN, USA Finetex Mats^TM^ by Finetex Technology Inc., USA and AMSOIL Ea Air Filters, USA and Canada [45].

Due to the limited mechanical and strength properties of the thin nanofiber layer (1 to 5 µm), it is applied on the surface made of a standard filter media with higher thickness and strength. The nanofibers can be arranged on one or both sides of the filter medium (cellulose, nylon or polyester). Using nanofibers as an additional layer on the filter media for air filters used in motor vehicles significantly increases the separation efficiency and filtration performance.

The designs of inlet air filters used in motor vehicle engines (Abrams M1 tank) using a filter element with added nanofibers and automatic pulse jet cleaning system PJCA (Pulse Jet Air Cleaner) are known from the literature [14,45]. This solution extends filter service life several times, and thus the time between filter replacements. PJAC system allows the filter to operate normally until the pressure drop at the filter exceeds a permissible value. After reaching a specific pressure drop, a pressure modulator is activated for 0.1 to 0.35 s to generate a compressed air pulse (0.4 to 0.6 MPa). The compressed air flowing in the opposite direction to the direction of air in the filtration process, removes the dust particles from the filter element surface; the dust particles are further removed to the dust collector [27,45].

Filter media of the cellulose type, made of relatively large fibers with a diameter in the range of 10–15 µm, which ensure air filtration accuracy above 2–5 µm, are commonly used for filtering the engine inlet air. As all dust grains above 1 µm wear the engine components, there is a need to improve the filtration efficiency of the engine intake air. The use of ultrafine fibers (less than 1 µm) has become the subject of interest in academic research and in industry, for example [14,46,47,48,49,50,51].

The authors of the work [14] presented an innovative filtering material made of sub-microfiber, which provides protection of engines against dust. A sub-microfiber substrate was prepared on a two-ply paper machine and its dust loading performance was compared with other filter materials in laboratory and field tests. The pressure drop of the used standard heavy duty (HD) filter after 1000 km of mileage and at a nominal flow rate of 1500 m^3^/h was over 2 kPa, which is the end of service life. Under the same conditions, the wet-folded submicrofiber filter achieved a pressure drop of more than 1 kPa, i.e., approximately 45% lower, demonstrating the advantages of the new filter material in the field test. In laboratory testing, when the pressure drop of a standard heavy duty medium reached 2 kPa, the pressure drop of the wet-laid submicrofiber material in the same bed of the dust mass was 38% lower than that of the standard heavy-duty medium.

In the work [46], cellulose nanofibers were prepared using the Lyocell fiber nanofibrillation method with the use of a PFI refiner, and the influence of nanofibers in the filter bed on the filtration efficiency and flow resistance was investigated. The mass fraction of nanofibers in the filter bed was changed in the range of 0–20%. With the increase in the proportion of nanofibers, the pressure drop increased. Upon reaching a share of 15%, the pressure drop increased sharply. For particles ranging in size from 0.03 to 2 µm, the separation efficiency curves show a typical V shape. With the increase in the proportion of nanofibers, the separation efficiency assumed higher and higher values. The particle size at which the performance was lowest on the performance curve was that of the Most Penetrating Particles Sizes (MPPS). The MPPS for all five cases was approximately 200 nm. The separation efficiency of MPPS increased from 0.1 to 0.6, indicating that cellulose nanofiber made from Lyocell was effective in improving the separation efficiency of the filter paper.

The authors of the work [47] developed a modified melt-blown technology which allowed us to produce filters composed of micrometer as well as nanometer sized fibers. One conventional microfibrous filter and five nanofibrous filters were examined. The complete structural characteristics, pressure drop and separation efficiency of removal of aerosol particles with diameters 10–500 nm were determined for all media.

The results of the experiments confirmed that using nanofibrous filters a significant growth of separation efficiency for the MPPS range can be achieved and the pressure drop rises moderately. Simultaneously, we noticed a shift of the MPPS towards smaller particles. Consequently, the quality factor for bilayer systems composed of a microfibrous support and a nanofibrous facial layer was considerably higher than this one for a conventional microfibrous filter alone.

Wang et al. [48] introduced novel hybrid filters of aligned carbon nanotube (CNT) sheets, sandwiched between electrospun polyimide (PI) nanofiber membranes serving as the supporting layers, were fabricated for the capture of fine particles. The CNT sheets and PI nanofiber membranes were thermally bonded together by melting electrospun polyetherimide (PEI) nanofibers. Two different kinds of filter structures were prepared, where multiple layers of aligned CNT sheets were either stacked together on top of each other or separated from each other by a PEI layer. The separation efficiency tests showed that the filtration efficiency increased with increasing number of CNT sheets. The maximum separation efficiency reached 99.99% at 0.053 m/s face velocity for 0.3-µm particles by the four-layer CNT filter, while the pressure drop was only 120 Pa.

Pei et al. [49] presented the results of research on the characteristics of the inlet air filter material covered with nanofibers obtained with the use of potassium chloride, ammonium sulfate, and ammonium nitrate particles in both dry and wet states. The performance was compared to that of conventional cellulosic filter media using 4 inches of water under various test conditions. The research was aimed at examining the effect of relative humidity and hygroscopic salt particles on the performance of an air filter covered with nanofibers. Switching from a conventional cellulose filter material to a nano-coated filter material can improve filter life by a factor of three or more depending on the material used in this test. However, when the particles are wet, the nanofiber layer reduces the volumetric load to about 50% based on the media used in this test.

Liu et al. [50] investigated the relationship between filtration efficiency, inlet velocity, and particle size of a PTFE-coated filter medium. The results of tests of three filters covered with a polytetrafluoroethylene (PTFE) membrane are presented. A number of experiments were carried out on filter performance and related parameters such as particle size and frontal velocity. In the experiments, 0.1% NaCl solution particles with sizes 10, 20, 50, 100, 200 and 300 nm were used. The average particle size was about 50 nm. The separation efficiency tests were carried out at inlet velocities of 0.003, 0.01, 0.053, 0.1 and 0.15 m/s. For particles from 10 to 300 nm, the separation efficiency curves show a typical V shape. The lowest point of the V-shaped curve is the minimum efficiency and corresponds to the Most Penetrating Particles Sizes (MPPS). At 0.053 m/s, for Filters A, B and C, the minimum separation efficiencies are 99.800%, 99.997% and 99.993%, respectively, and the MPPS values are 100, 70 and 100 nm, respectively. Membranefilters with larger pore sizes allow more penetration of large particles.

Liu et al. [51] presented studies that aimed to determine the effect of pore size and fiber diameter on the performance of ultra-low emission bag filters. In this study, three kinds of conventional polyester filter (depth filtration media) and two kinds of polytetrafluoroethylene membrane-coated polyester filter (surface filtration media), having various filter pore sizes and fiber diameters, were tested to determine the performance of static and dynamic filtration. In the dynamic filtration performance experiments, 50% of the test dust was less than 2.5 µm in size, and the mass mean aerodynamic diameter of the dust was 1.5 µm. The filtration velocity was 2 m/min (0.033 m/s), and the dust concentration was *s* = 18.4 g/m^3^. In the depth filtration media, the separation efficiency and the pressure drop of the fabric structure were improved when the filter pore size and the fiber diameter were smaller in magnitude.

The filter media must be evaluated for separation efficiency, filtration performance and changes in pressure drop. The manufacturers of filter media often specify the structural parameters (pore size, air permeability, thickness) without giving any information on the filtration performance for a test dust. This data can be obtained by experimental tests on filter medium specimens that are costly and time consuming; however, those test methods are the most reliable. The study presents methods and test results for standard filter media (cellulose and polyester) and standard filter media with a nanofiber layer and PTFE membrane. The results can be used to choose the most suitable filter medium for the inlet air of a motor vehicle’s engine to minimize the engine wear and extend its mileage.

## 2. Authors’ Own Research

Filter material experimental tests included organization, research and results analysis. In the organizational part, a test stand and particle counter were prepared, and test conditions were adopted: dust concentration, *Q_w_* stream range, filtration speed, and measurement duration cycle. A research methodology was developed. Filter material samples (filter cartridges) were selected for testing and they were marked as A, B, C, D and E. Filter materials data were summarized in the table.

The research part was carried out in two stages, with the following courses:Flow characteristics Δ*p_w_* = *f*(*Q_w_*) of selected filter materials (cartridges) were made by making five repetitions for each *Q_w_* value.Efficiency *φ_w_*, filtration accuracy *d_zmax_* and flow resistance Δ*p_w_* characteristics depending on dust absorption coefficient *k_m_* of three filter cartridges of each material, A, B, C, D and E, were made.


Measurement result analysis was performed in the scope of:From each material, A, B, C, D and E, one of the sample characteristics were selected in order to compare (on one graph) their initial efficiencies and duration of the initial period, as well as changes in flow resistance depending on dust absorption coefficient *k_m_*.On the example of cartridge A (cellulose), changes in the number of dust grains in the air behind the filter cartridges in successive measuring intervals and after subsequent test cycles are presented.On the example of filter cartridge A and E, the particle size distribution of grains in the exhaust air was analyzed after measurement No. 1 and after measurement corresponding to the end of the initial filtration period.Using the quality factor *q*, the filtration properties of the tested material samples were assessed.

Results analysis was completed with a summary in the form of conclusions.

### 2.1. Purpose, Scope and Subject of Tests

The tests aimed to determine and compare the filtration parameters including separation efficiency, filtration performance and pressure drop at the filter elements made of different filter media (cellulose, polyester, cellulose and polyester, cellulose and polyester with nanofiber layer, polyester with PTFE membrane) by determining the following filtration characteristics:Filtration performance *d_zmax_* = *f*(*k_m_*);Separation efficiency *φ_w_ = f*(*k_m_*);Pressure drop Δ*p_w_ = f*(*k_m_*);Flow (aerodynamic) characteristics Δ*p_w_ = f*(*Q_w_*);
where *Q_w_* (m^3^/h) is the air volume flow rate through the filter element, *k_m_* is the mass loading of dust *m_z_* retained and evenly distributed over 1 m^2^ of the active filter media surface area, expressed as:(1)$$k_{m}=\frac{m_{z}}{A_{w}}\left(\frac{g}{m^{2}}\right).$$

The filtration rate is defined as a product of air volume flow rate through the filter element *Q_wmax_*, equal to the engine air demand at maximum power *n_N_* and the active surface area of the filter paper *A_w_* based on the following relationship:(2)$$v_{Fw}=\frac{Q_{w}}{A_{w}\times 3600}\left(m/s\right).$$

The tests covered the filter elements (Figure 1) of the same type, with identical dimensions and filtration surface area *A_w_* = 0.1534 m^2^ made of different filter media. To facilitate the result analysis, the filter media (elements) has been assigned a letter designation, A, B, C, D and E.

Table 1 shows characteristic parameters of the tested filter media. A nanofiber layer is applied on the inlet side of the filter media D (cellulose+polyester). A microporous PTFE membrane is applied on the inlet side of the filter media E (polyester). The filter media E shows 10 times lower air permeability compared to the filter media A and is 2 times lower compared to the other tested filter media.

### 2.2. Test Methods and Conditions

Tests were carried out on a test stand (Figure 2) with a Pamas 2132 particle counter with HCB-LD-2A-2000-1 sensor. The particle counter records the number and size of dust particles in the air *Q_w_* downstream of the filter element from 0.7 to 100 µm at *i* = 32 intervals by diameter (*d_zimin_*–*d_zimax_*).

At a specific distance downstream of the filter, a tip of the measuring probe is installed centrally in the axis of the measuring line supplying the air to the particle counter. A special filter protecting the rotameter is installed at the measuring line. A PTC-D test dust, an equivalent of AC-fine test dust was used. Figure 3 shows chemical composition and particle size distribution of test dust [48]. The mass fraction of 0- to 5-µm particles in the overall dust mass is nearly 40%. Due to its small size, the dust particles are not easily retained by the porous filter media. Over 67% are SiO_2_ particles—a mineral showing high hardness (7 out of 10 on a Mohs scale), causing engine components wear.

Flow characteristics Δ*p_w_ = f*(*Q_w_*) of the filter elements were determined at 8 points for the air volume flow rate *Q_w_* = *Q_wmin_*–*Q_wmax_*. The maximum air volume flow rate *Q_wmax_* was determined for the maximum filtration rate *υ_Fw_* = 0.1 m/s. For passenger car filters, the maximum filtration rate of the filter paper is between *υ_Fw_* = 0.07 to 0.12 m/s [18,20,53,54,55,56]. For the filtration rate of (*υ_Fw_ =* 0.1 m/s), the maximum air volume flow rate calculated from the following relationship is *Q_wmax_* = 56 m^3^/h.
(3)$$Q_{wmax}=A_{w}\;\cdot v_{Fw\;}\cdot 3600\left(m^{3}/h\right).$$

A RIN 60 type rotameter with a measuring range of 3 to 67 m^3^/h and accuracy class of 2.5 was used to measure the air volume flow rate *Q_w_*.

Filtration characteristics of the filter elements: separation efficiency *φ_w_ = f*(*k_m_*), filtration performance *d_zmax_ = f*(*k_m_*) and pressure drop Δ*p_w_ = f*(*k_m_*) were determined using a gravimetric method at a constant filtration rate *υ_Fw_* = 0.1 m/s. Dust mass retained by the tested filter element and the absolute filter in subsequent j cycles after time τ*_p_* (evenly dosing time) to the filter was determined. Dust concentration at the filter element inlet was *s* = 0.5 g/m^3^. The duration (uniform dust batching time) was τ*_p_* = 3 min in the initial period (I) and τ*_p_* = 9–12 min in the main period (II) of the filter working time. After each cycle *j*, the following parameters were determined for the filter element: separation efficiency, filtration performance, pressure drop and dust absorption coefficient. Dust mass retained by the filter element and the absolute filter was determined using an analytical balance with a 220-g capacity and a 0.1-mg readability.

During the measuring cycle, the particle counter was activated 60 s before the planned test end to count the number and size of dust particles in the air downstream of the filter.

Following points were determined after each measuring cycle *j*:

Pressure drop Δ*p_wj_* at the filter as a static pressure drop upstream and downstream of the filter based on the measured height Δ*h_mj_* (after the dust supply ended) at a U-tube manometer using the following relationship:(4)$$\Delta p_{wj}=\frac{\Delta h_{mj}}{1000}\cdot \left(\rho_{m}-\rho_{H}\right)\cdot g\left(\mathrm{Pa}\right),\;$$
where ρ*_m_* is the manometric liquid density (kg/m^3^), ρ*_H_* is the air density (kg/m^3^), *g* is the gravitational acceleration (m/s^2^).
Separation efficiency, as a dust mass product *m_ZFj_* retained by the filter and dust mass *m_Dj_* supplied to the filter during the next *j* cycle based on the following relationship:(5)$$\varphi_{j}=\frac{m_{Fj}}{m_{Dj}}=\frac{m_{Fj}}{m_{Fj}+m_{Aj}}100\%.$$Dust mass loading *k_mj_* of the tested filter medium:(6)$$k_{mj}=\frac{\sum_{j=1}^{n}m_{Fj}}{A_{w}}\;(g/\;m^{2}).$$Number of dust particles *N_zi_* in the air stream downstream of the filter (passing through the filter media) in the intervals by diameter (*d_zimin_* to *d_zimax_*).Filtration performance as the largest dust particle size *d_zj_* = *d_zmax_* in the air downstream of the filter.Fraction of each dust particle size in the air downstream of the filter for each cycle:(7)$$U_{zi}=\frac{N_{zi}}{N_{z}}=\frac{N_{zi}}{\sum_{i=1}^{32}N_{zi}}100\%,$$
where $N_{z}=\overset{32}{\underset{i=1}{\sum }}N_{zi}$–the total number of dust particles passing through the filter (for all measuring intervals) in the test cycle.


Test cycle used in the study involved five counts of dust particles from the range of 0.7 to 35 μm, divided into 32 intervals by diameter (*d_zmin_* to *d_zmax_*).

In accordance with the method, the following filtration characteristics were determined: separation efficiency *φ_w_ = f*(*k_m_*), filtration performance *d_zmax_ = f*(*k_m_*) and pressure drop Δ*p_w_ = f*(*k_m_*) of the filter elements A, B, C, D and E. Three specimens of each filter element with the same filter medium were tested. The flow characteristics Δ*p_w_ = f*(*Q_w_*) were determined before the filter element tests using the test dust.

## 3. Test Result Analysis

Figure 4 shows the test results of the flow characteristics Δ*p_w_* = *f*(*Q_w_*) of the tested filter elements. A parabolic increase in pressure drop Δ*p_w_* = *f*(*k_m_*) with an increase in air volume flow rate can be observed, corresponding to the data from the literature.

The highest pressure drop in the entire tested air volume flow rate *Q_w_* range was observed for the filter element E—a filter bed made of polyester with PTFE membrane. For *Q_wmax_* = 56 m^3^/h, the pressure drop at the filter element E is ∆*p_w_* = 528 Pa (Figure 4). It is over 20% more than the pressure drop at the polyester bed and over 70% more than the pressure drop at the filter medium A (cellulose). The filter medium A achieves the lowest pressure drop (∆*p_w_* = 312 Pa) (Figure 4), due to high permeability (838 dm^3^/m^2^/s), significantly higher than observed for other filter media. The pressure drop at the other filter media (C and D) is between 370 and 410 Pa. The significantly higher pressure drop at the filter medium E compared to other filter media is due to a very low permeability of the filter bed (80 dm^3^/m^2^/s)—see Table 1.

The example test results for the filtration characteristics including separation efficiency *φ_w_ = f*(*k_m_*), filtration performance *d_zmax_ = f*(*k_m_*) and pressure drop Δ*p_w_ = f*(*k_m_*) of three specimens of filter elements A and D and three specimens of filter element E are shown in Figure 5 (element A), Figure 6 (element D) and Figure 7 (element E).

The presented separation efficiency *φ_w_*, filtration performance *d_zmax_* and pressure drop Δ*p_w_* of three filter element A specimens (Figure 5) differ slightly (form and value). The curve is similar and corresponds to the information provided by the authors of other research studies [20,22,56]. However, there are differences in the values of the initial filtration efficiency (for dust mass loading from 0 to 70 g/m^2^) for three A filter cartridges (Figure 5). According to the authors, the different filtration efficiency values obtained at the initial stage result from the different structure of the same type of paper in individual cartridges. The active surface of the paper in each filter could also have been different (than assumed on the basis of the filter producer’s data).

This had a direct impact on the effectiveness of the inertia and direct hooking mechanism, which are decisive in depth filtration in the bed. The authors assumed that all filter cartridges are similar which is related to their mass production. In particular, it was assumed that the filter material structure and its thickness in each filter cartridge is the same. It was also assumed that the active surface area of the cartridges is the same. With the same value of the air stream Q during the tests, the filtration speed *υ_Fw_* should also be the same.

Similar curves can be observed for the filter elements D (Figure 6) and E (Figure 7). Significant differences in the form and values of curves for the filter element A, D and E are due to the different filter media and different structural parameters.

Figure 8 and Figure 9 show separation efficiency *φ_w_ = f*(*k_m_*), filtration performance *d_zmax_ = f*(*k_m_*) and pressure drop Δ*p_w_ = f*(*k_m_*) of the filter elements made of standard filter media: A (cellulose) and B (polyester). The characteristics were compared with the characteristics of the filter elements with a filter media being a composite of different layers: C (cellulose+polyester), D (cellulose+polyester+nanofibers), E (polyester+PTFE membrane).

Separation efficiency *φ_w_*, filtration performance *d_zmax_* and pressure drop Δ*p_w_* of one out of three filter element A, B, C, D and E specimens were used in the analysis. The characteristics of five filter elements A, B, C, D and E with different filter media have a similar form and significantly different values.

With an increase in dust mass retained by the filter layer (increase in *k_m_* factor), separation efficiency, filtration performance and pressure drop of the filter elements A, B, C, D and E increase, however, at a different rate and with different initial values. The characteristics are the result of the dust particles being retained by the fiber surface and filling the voids (pores). This phenomenon has been widely discussed in the literature [20,22,38,41].

By convention, the operation of the tested filter elements can be divided into two stages. The first (I), initial stage of operation of each filter element lasts until the separation efficiency of *φ_w_* = 99.9% is reached. This stage is characterized by a low initial separation efficiency, filtration performance and pressure drop.

Initial separation efficiencies of the tested filter elements vary. The lowest value (*φ_w0A_* = 96.3%) was observed for the filter element made of filter medium A (cellulose). The elements B, C, D and E show higher initial separation efficiency: *φ_w0B_* = 98/9%, *φ_w0C_* = 98.2%, *φ_w0D_* = 99.8%, *φ_w0E_* = 99.97%, respectively (Figure 8).

With an increase in dust mass retained by the filtration layer (increase in *k_m_* factor), the separation efficiency of the tested filter elements increases. The required separation efficiency (*φ_w_* = 99.9%—end of stage I) is achieved by the filter elements working in similar conditions (the same dust concentration and air volume flow rate) at different times. Filter element A reaches efficiency of *φ_w_* = 99.9% at the mass loading of dust of *k_mA_* = 110.7 g/m^2^. For the elements B, C, D and E made of different filter media, the first stage is significantly shorter.

For the filter element E (polyester+PTFE membrane) and the filter element D (cellulose+polyester+nanofibers), stage I ends first after reaching the mass loading of dust of *k_mE_* = 5.77 g/m^2^ and *k_mD_* = 7.22 g/m^2^, respectively. For the filter element B (polyester), stage I ends at the mass loading of dust *k_mB_* = 31.9 g/m^2^, and for the filter element C (cellulose + polyester), at the mass loading of dust *k_mC_* = 48.5 g/m^2^ (Figure 7).

At the beginning of the filtration process, the maximum size of dust particles downstream of the filter (in the filtered air) may vary. The largest dust particles (*d_zmax_* = 28 µm) were observed downstream of the filter element A (cellulose), and the smallest (*d_zmax_* = 7.9 µm), downstream of the filter element E which is strictly correlated with the initial separation efficiency of the filter elements. This results from the higher basis weight and smaller pores of the tested filter media (Table 1).

With an increase in dust mass retained by the filter bed, the particles accumulate at the surface of the porous structure fibers and at the surface of previously accumulated particles, creating a constantly growing and complex dendritic structures (agglomerates), filling the voids between the fibers. The result of changes in the bed structure is an increase in separation efficiency. The size of the dust particles downstream of the filter decreases gradually. After I stage, the dust particles are between *d_zmax_* = 10.3–11.9 µm (filter element A) and *d_zmax_* = 3.1–5.5 µm (filter element D and E).

The initial stage of operation of filter element A made of cellulose is several times longer than observed for other filter elements. It significantly affects the durability of the engine. Low separation efficiency and filtration performance in the initial stage (after using a new filter element) results in the presence of large dust particles above 5 µm, significantly affecting the premature wear of engine components, mostly piston, piston rings and cylinder sleeves. The duration of the initial filtration stage may be reduced by using the filter medium with a layer of nanofibers or a PTFE membrane. High separation efficiency and filtration performance 99.9% is achieved by the filter elements much earlier than in filters made of standard filter media (cellulose) to eliminate the effect of air impurities on premature wear and durability of the engine.

The filtered air contains dust particles with different sizes and in different volumes. A predominant particle size is between 0.7 and 1.1 µm (Figure 9). With an increase in dust particle size *d_z_* in the air downstream of the filter, the number of particles *N_p_* is reduced, indicating the increasing separation efficiency of the tested filter medium. In the last interval, a single dust particle is usually present (*d_z_* = *d_zmax_* maximum size) (Figure 9) that is used as a measure of filtration performance.

Figure 10 shows the dust particle size distribution (test dust) upstream and downstream of the filter element A (cellulose). Dust with a particle size of up to 80 µm is observed in the filter inlet air. A predominant particle size (*U_pt_* = 16.2%) in the dust is 4 µm.

The size distribution of dust particles in the air downstream of the filter is completely different. The largest fraction of dust particles *U_p_* in the air downstream of the filter are the dust particles with size between *d_z_* = 0.7 and 1.1 µm (Figure 10). For the filter element A (measurement no. 1–*k_m_* = 2.68 g/m^2^) the fraction is *U_p_*_1_ = 26.3%. For subsequent measurements, the fraction increases and for measurement no. 11 (*k_m_* = 110.7 g/m^2^) is *U_p_*_11_ = 77.5%.

The fraction of particles significantly decreases with an increase in dust particle size and the filtration process duration. The fraction of dust particles *d_z_* = 2 µm for measurement no. 1 is *U_p_*_1_ = 11.6%. For subsequent measurements, the fraction of particles *d_z_* = 2 µm increases and for measurement no. 11 is *U_p_*_11_ = 1.5% (Figure 10). A similar change in the share of *U_p_* of dust grains in subsequent measurement intervals in the air behind the filter element occurs for other tested materials A, B, C, D and E. There are differences in the values of the *U_p_* shares. The shorter the initial period of filtration, the greater the proportion of grains with dimensions *d_z_* = 0.7–1.1 µm (after measurement No. 1). In the air behind the E insert (polyester+PTFE membrane) for measurement no. 1 (*k_m_* = 5.77 g/m^2^), the share of grains with dimensions *d_z_* = 0.7–1.1 µm has the value *U_p_*_1_ = 67.9%. For subsequent measurements, the share of these grains increases less intensively and does not exceed the value of *U_p_* = 70%.

Figure 11 and Figure 12 show the detailed particle size distributions in the air downstream of the filter element A after reaching the mass loading of dust *k_mA_* = 2.68 g/m^2^ (measurement no. 1) and *k_mD_* = 110.7 g/m^2^ (measurement no. 11).

Figure 13 shows the particle size distribution in the air downstream of the filter element E after reaching the mass loading of dust *k_mA_* = 5.77 g/m^2^ (measurement no. 1). The fraction of dust particles *d_z_* = 0.7 to 1.1 µm is *U_p_*_1_ = 67.9%. It is similar to the value recorded for the filter element A at the end of the initial stage.

The accumulation of dust particles on the panel filter fibers changes the filter structure, reduces the surface area of the air stream flow around the fibers. The aerosol flow rate increases with an increase in pressure drop at the filter bed. An increase in pressure drop at the tested filter elements, despite similar testing conditions, differed in rate (Figure 14). The pressure drop of Δ*p_w_* = 3 kPa is achieved by the filter elements after retaining different masses of dust.

Filter elements with a nanofiber layer are characterized by the lowest dust mass loading. It is determined by the surface filtration during which the dust particles are retained by the nanofiber layer before reaching the inside of the filter bed. At the point, the filter elements reach the pressure drop of Δ*p_maxE_* = 3 kPa, they retain and accumulate different masses of dust. Filter elements A and D reach the dust mass loading of *k_mA_* = 210 g/m^2^ and *k_mE_* = 108 g/m^2^, respectively (Figure 12). Dust mass Δ*k_m_* retained by the filter elements, from the point of achieving the separation efficiency of *φ_w_* = 99.9% until the point of achieving the pressure drop of Δ*p_maxE_* = 3.0 kPa is similar and is approx. 100 g/m^2^ (Figure 12). It is determined by the duration of the initial period (I), which lasts from the moment the filter element achieves the separation efficiency of *φ_w_* = 99.9%. In this period, the air downstream of the filter contained large dust particles (*d_zmaxA_* = 28–15 µm), affecting the engine durability. For filter element (A), the initial stage (I) is over 50% of its total service life, and significantly affects the abrasive wear of engine components. For filter elements (C, B and D), the period is shorter and is 40%, 22% and 6% of its total service life, respectively.

In the second (II) stage of the operation of the filter elements B, C, D, the separation efficiency is maintained at a stable level of *φ_w_* = 99.9%. For the filter element A, a slight and systematic decrease in separation efficiency can be observed. In the last measurement, the separation efficiency was *φ_w_* = 99.7%. It may be due to the accumulation of a significant amount of dust in form of extensive tree-like dendrites [57]. The dust particles at the top of the dendritic structures are captured and transferred to the outlet side of the filter medium, and captured by the air stream from the engine cylinders. The decrease in the separation efficiency is related with the maximum dust particle size, reaching higher and higher values. For the last measurement, the value was *d_zmaxA_* = 16.7 µm. For the remaining filter elements, the maximum dust particle sizes *d_zmax_* were stable at a significantly lower level. Downstream of filter element D with a layer of nanofibers, the particle size was *d_zmaxD_* = 3.1 to 5.1 µm.

The filtration properties of the tested filter media were evaluated using a quality factor *q* [31].
(8)$$q=\frac{-\ln\left(1-\frac{\varphi_{0}}{100}\right)}{\Delta p}\left(\frac{1}{\mathrm{kPa}}\right),$$
where *φ*_0_–initial efficiency of the filter bed [%], ∆*p*–pressure drop for nominal air volume flow rate (kPa).

The calculation results shown in Figure 15 indicate that the highest quality factor was observed for specimen E (polyester+PTFE membrane) and specimen D: *q_E_* = 15.4 kPa^−1^ and *q_D_* = 15.1 kPa^−1^, respectively. Specimen E includes a membrane and specimen D includes a nanofiber layer. The specimens reach higher initial values of the separation efficiency: *φ_owE_* = 99.97% and *φ_owD_* = 99.8%, and filtration performance in the initial stage *d_zmaxE_* below 15.8 µm and below *d_zmaxD_* 7.9 µm, respectively.

A significantly lower value of the quality factor *q* = 10.6 kPa^−1^ is achieved by specimen A (cellulose) as a result of lower initial separation efficiency *φ*_0*wA*_ = 96.3%. The separation efficiency is correlated with the filtration performance. Lower separation efficiency means not only lower mass of dust retained by the filter, but also larger dust particles in the air downstream of the filter, affecting the wear and durability of the internal combustion engine components.

## 4. Conclusions

A characteristic feature of the aerosol separation process in the filter media is an initial period characterized by low separation efficiency *φ_w_*, filtration performance *d_zmax_* and pressure drop Δ*p_w_*. The air downstream of the filter element may contain large dust particles (28 µm), causing premature wear of engine components, in particular pistons, piston rings and cylinder sleeves, affecting its performance and durability.The initial filtration period (time until the required filtration efficiency is reached-*φ* = 99.9%) varies and depends on the filter medium used. A nanofiber layer or a PTFE membrane applied on the substrate made of conventional filter media (cellulose, polyester) significantly increase the separation efficiency *φ_w_* and filtration performance *d_zmax_*. The initial filtration stage is several times shorter than for the filter element made of standard filter media only, which significantly reduces the wear of engine components.With an increase in dust mass retained by the filter element (with an increase in dust mass loading *k_m_*) the filtration performance and separation efficiency of the tested filters and pressure drop increase. Filter elements with a nanofiber layer and a PTFE layer show twice the rate of increase in pressure drop Δ*p_w_* compared to the filter elements without those layers. The service life (vehicle mileage) until the permissible pressure drop is reached will be shorter, requiring more frequent filter element replacements, on one hand increasing the operating costs, on the other, due to higher filtration performance and reduced wear of engine components, extending the mean time to repair. The presented phenomenon of interrelationship increase in efficiency and accuracy of inlet air filtration with a significant increase in flow resistance applies to all filter materials with nanofiber layers. Data of bed with a PTFE layer are based only on the results obtained by the authors during their own research.After reaching the permissible pressure drop Δ*p_wdop_* = 3 kPa, the filter elements with a composite filter media retain *k_m_* = 108–135 g/m^2^ of dust. For the same pressure drop Δ*p_wdop_*, filter element A (cellulose) achieves dust mass loading *k_m_* = 209 g/m^2^, almost 100% higher. The filter elements with a nanofiber layer or a PTFE membrane retain the dust at the surface, blocking the air flow through the layer, increasing the rate of an increase in pressure drop. It shows that in the filter beds with a nanofiber layer, a surface filtration process takes place, instead of a depth filtration process. Those type of filter media can be used in the filtration systems with pulse jet cleaning feature, significantly increasing the service life of the filter element.The results of the experimental study partially fill the gap in the field of basic properties of materials used in the design and selection of panel filters for the inlet air of the internal combustion engine.

Filter material experimental research, which aims to determine filtration efficiency, accuracy and flow resistance changes, consist in performing a large number of measurements with test dust usage. Many conditions must be met. The filtration efficiency is most often determined by mass method, in measuring cycles which last few or several minutes. By weighting with a high-accuracy analytical scale, the sample weight is determined before and after the test (dust dosing) and then compared with dust weight delivered at that time. Dust dosing during the measurement should be even and ensure a fixed dust concentration value, which is a big problem in cases where there is a lack of an appropriate dosing device. The air stream flowing through the filter material sample should be constant, which ensures a constant, defined filtration speed. During the test, dust is retained on the sample, which increases its flow resistance, which causes a decrease in the air stream. Therefore, air flow should be constantly monitored and adjusted during the measurement. The test stand is also equipped with an absolute filter that stops contaminants that have passed through the tested sample. It thus protects air flow meter measuring element. However, it increases flow resistance in the measuring system. Type of test dust used is important during the tests. Test dusts differ in their chemical and particle size composition and should be selected depending on the type of the intended ground on which the vehicle will travel. The station is equipped with a particle counter that records dust grains number and size in the air behind the filter. It is important to maintain the isokinetic condition of the probe collecting the polluted air stream for analysis. Tests are time consuming and require many conditions to be met before and during the measurement. Appropriately trained personnel are essential. Experimental research is expensive, but it is the most reliable research method.

## Figures and Tables

**Figure 1 materials-13-03498-f001:**
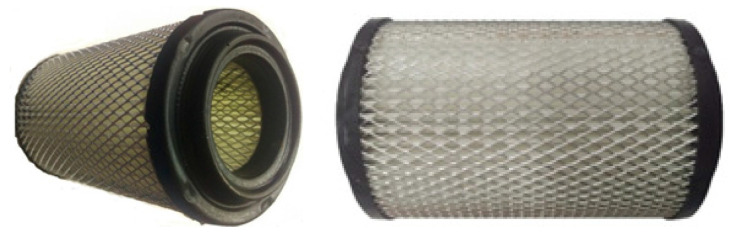
Filter element.

**Figure 2 materials-13-03498-f002:**
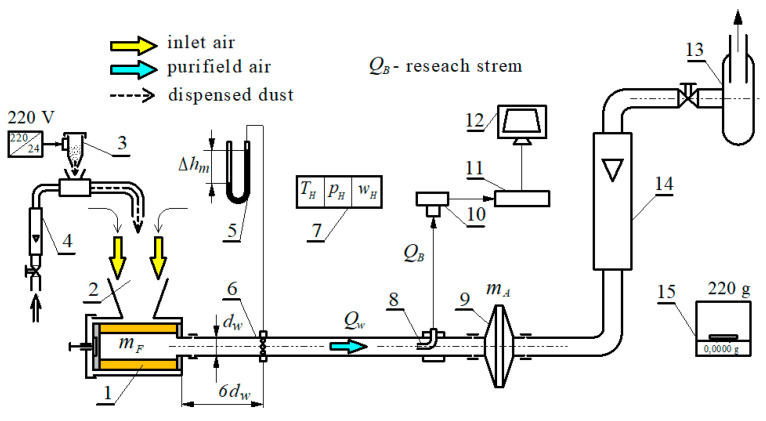
Diagram of the filter element test stand: 1–filter element, 2—dust chamber, 3—dust dispenser, 4—rotameter, 5—U-tube manometer, 6—measuring tube, 7—humidity, ambient air temperature and pressure measurement unit, 8—measuring probe, 9—absolute filter, 10—sensor, 11—particle counter, 12—measuring computer, 13—suction fan, 14—rotameter, 15—analytical balance.

**Figure 3 materials-13-03498-f003:**
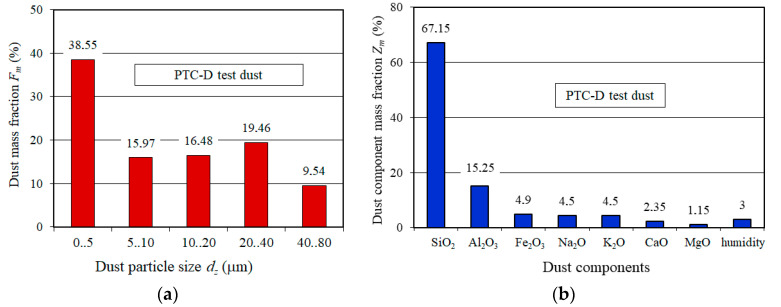
PTC-D test dust: (**a**) dust mass fraction, (**b**) dust component mass fraction [52].

**Figure 4 materials-13-03498-f004:**
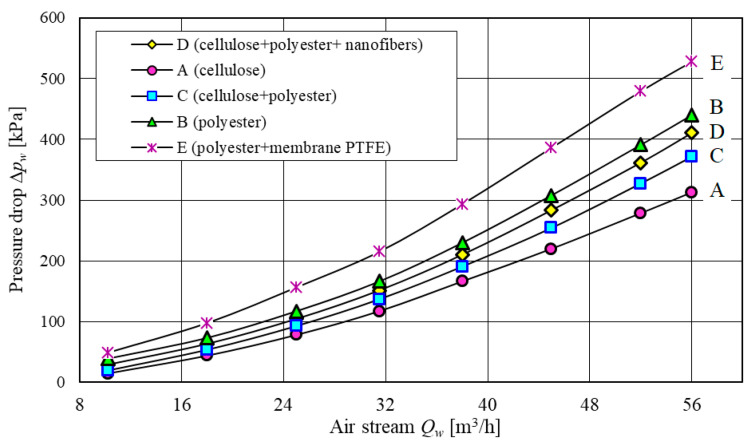
Flow characteristics Δ*p_w_* = *f*(*Q_w_*) of the tested filter elements.

**Figure 5 materials-13-03498-f005:**
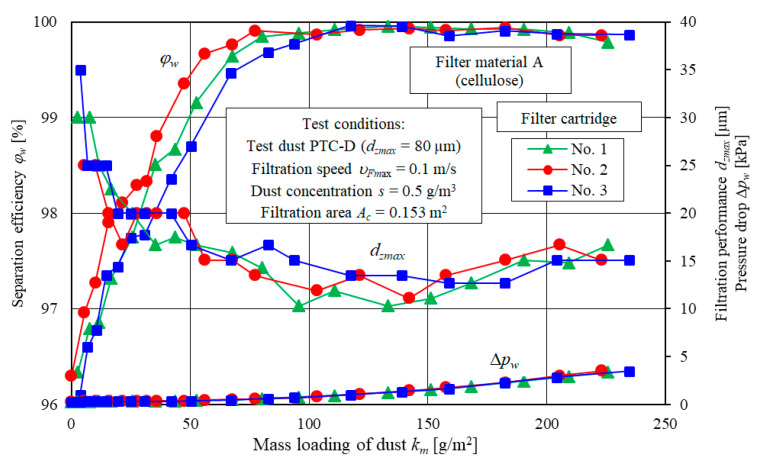
Separation efficiency *φ_w_*, filtration performance *d_zmax_* and pressure drop Δ*p_w_* as a function of dust mass loading *k_m_* for three filter elements A specimens.

**Figure 6 materials-13-03498-f006:**
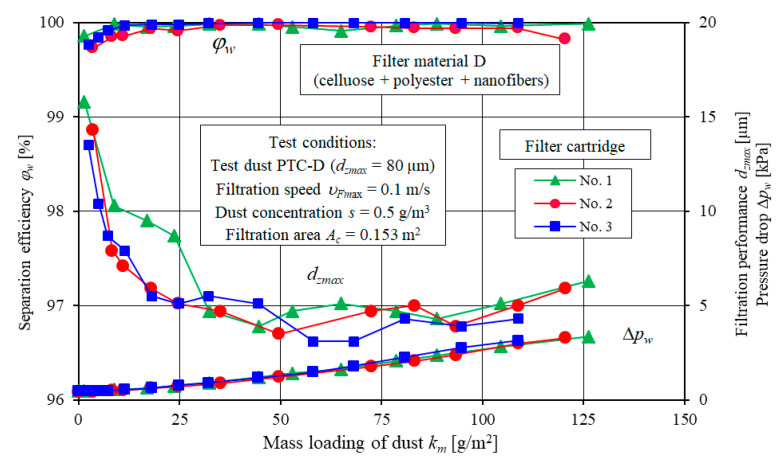
Separation efficiency *φ_w_*, filtration performance *d_zmax_* and pressure drop Δ*p_w_* as a function of dust mass loading *k_m_* for three filter elements D specimens.

**Figure 7 materials-13-03498-f007:**
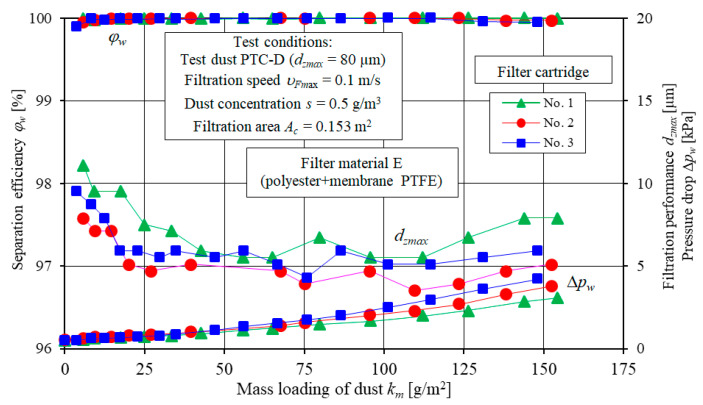
Separation efficiency *φ_w_*, filtration performance *d_zmax_* and pressure drop Δ*p_w_* as a function of dust mass loading *k_m_* for three filter elements E specimens.

**Figure 8 materials-13-03498-f008:**
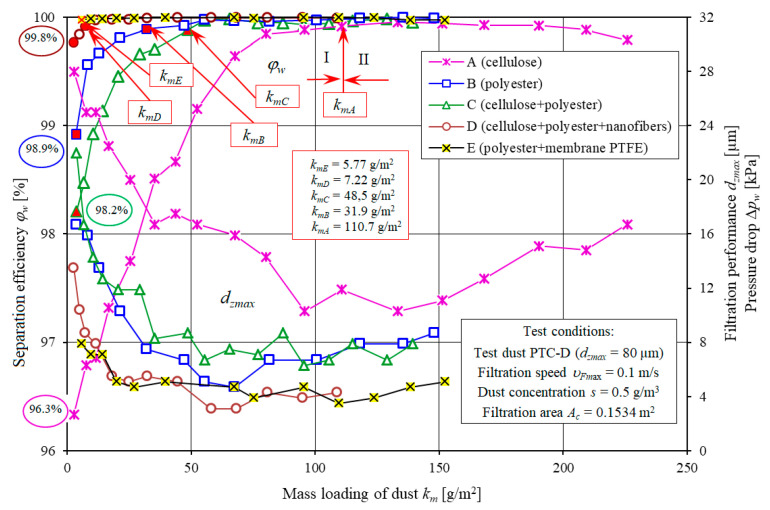
Separation efficiency *φ_w_* and filtration performance *d_zmax_* as a function of dust mass loading *k_m_* of the tested filter elements.

**Figure 9 materials-13-03498-f009:**
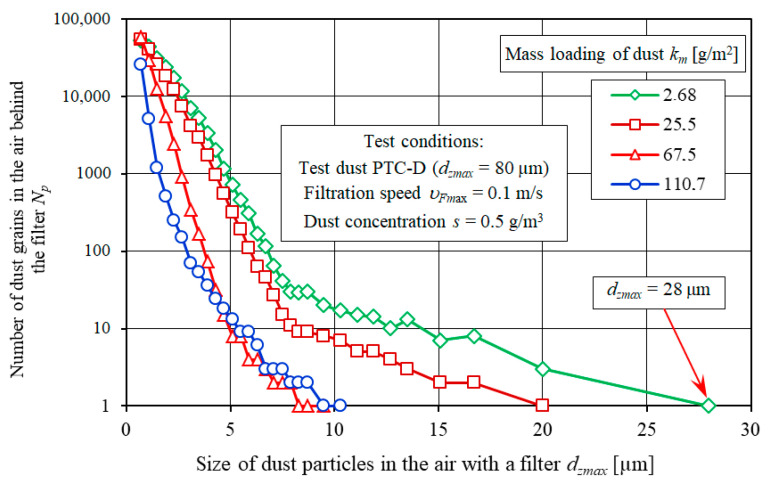
The number of dust particles in subsequent measuring intervals in the air downstream of the filter element A (cellulose).

**Figure 10 materials-13-03498-f010:**
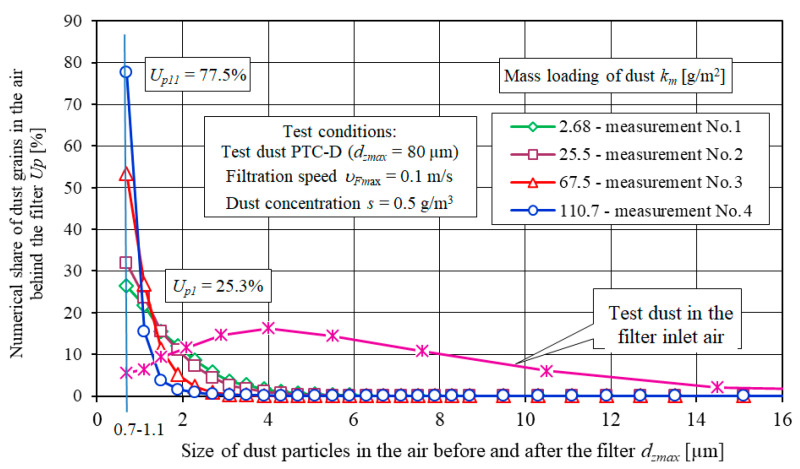
Share of dust particles in subsequent measuring intervals in the air upstream and downstream of the filter element A (cellulose).

**Figure 11 materials-13-03498-f011:**
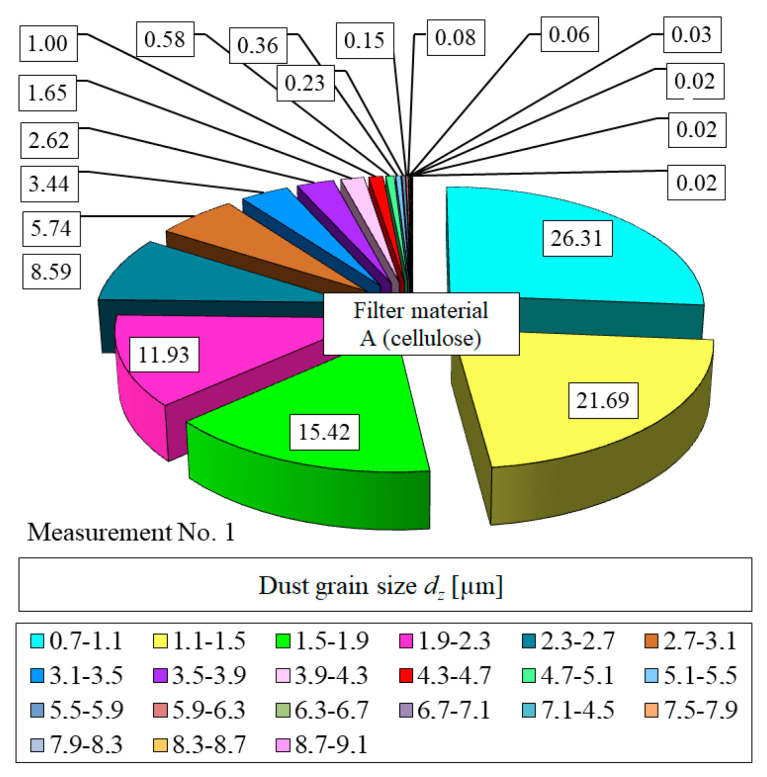
Particle size distribution in the air downstream of the filter element A after reaching the dust mass loading *k_mA_* = 2.68 g/m^2^ (measurement no. 1).

**Figure 12 materials-13-03498-f012:**
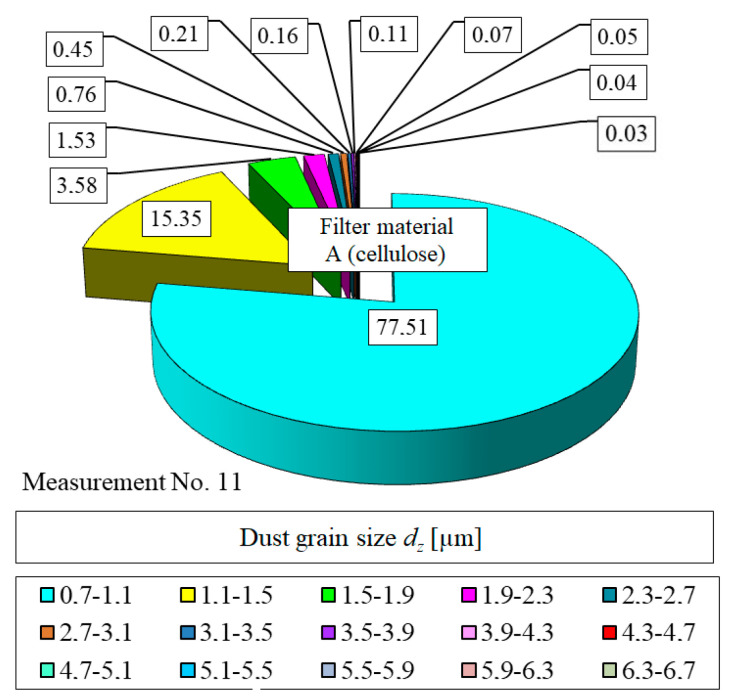
Particle size distribution in the air downstream of the filter element A after reaching the dust mass loading *k_mA_* = 110.7 g/m^2^ (measurement no. 11).

**Figure 13 materials-13-03498-f013:**
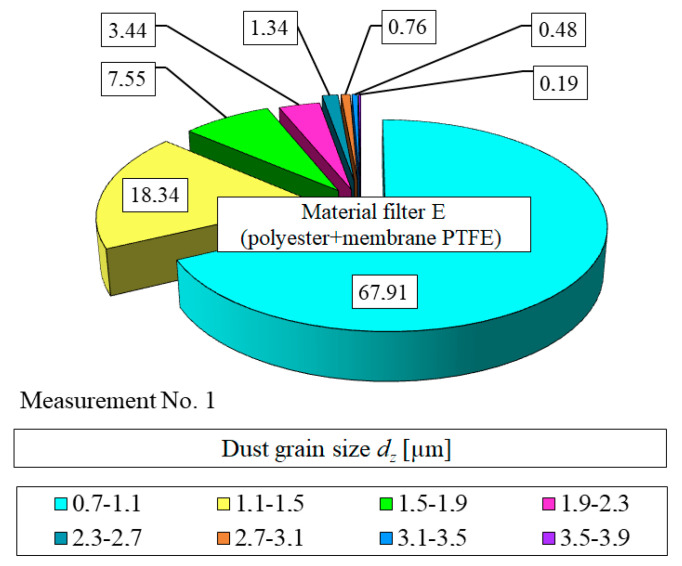
Particle size distribution in the air downstream of the filter element E after reaching the dust mass loading *k_mA_* = 5.77 g/m^2^ (measurement no. 1).

**Figure 14 materials-13-03498-f014:**
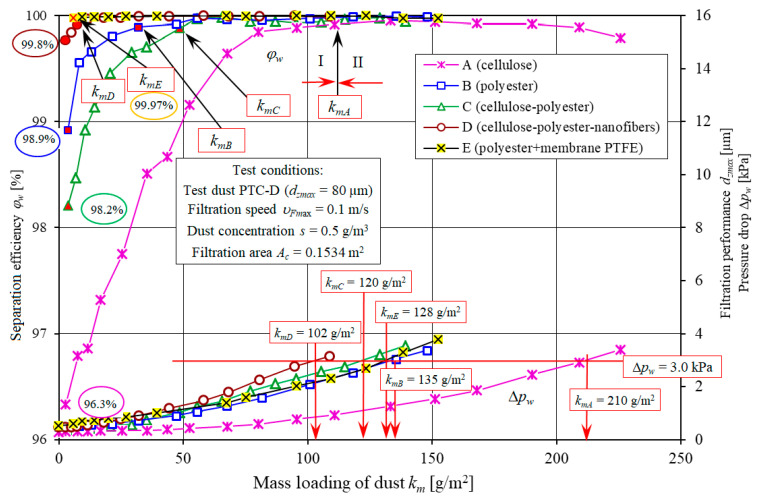
Separation efficiency *φ_w_* and pressure drop *∆p_w_* as a function of the mass loading of dust *k_m_* of the tested filter elements A, B, C, D and E.

**Figure 15 materials-13-03498-f015:**
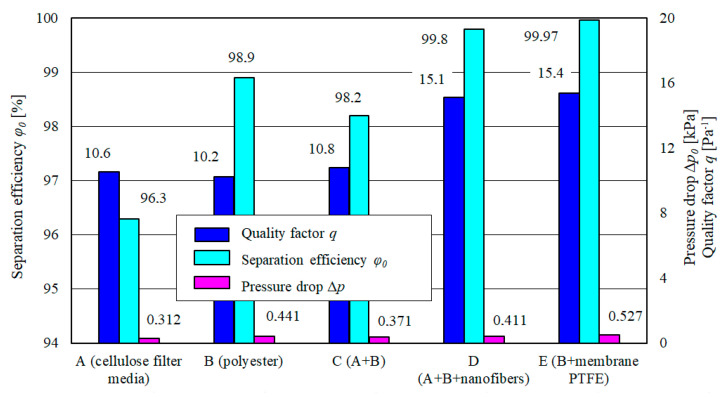
Separation efficiency of the tested filter media A, B, C, D and E.

**Table 1 materials-13-03498-t001:** The parameters of the tested filter media based on the data provided by the manufacturer.

Parameters	Filter Medium Identification				
	A	B	C	D	E
Filter medium	Cellulose	Polyester	90% Cellulose +polyester	Cellulose + polyester +nanofibers	Polyester + membrane PTFE
Permeability *q_p_* (m^3^/m^2^/h) at 200 Pa	3015	540	685	540	288
Basis weight *g_m_* (g/m^2^)	121	180	135	130	260
Thickness *g_z_* (µm)	610	550	360	300	560

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
