# Peer review of "Experimental Study of Filtration Materials Used in the Car Air Intake"

_materials, 2020, doi:10.3390/ma13163498_

Round 1

Reviewer 1 Report

In this manuscript entitled " Experimental study of materials for filtration of the intake air of a car engine ", the authors elucidated a study to choose the most suitable filter medium for the inlet air of a motor vehicle’s engine to minimise the engine wear and extend its mileage. As the authors mentioned, this results of the experimental study partially fill the gap in the field of basic properties of materials used in the design and selection of panel filters for the inlet air of the internal combustion engine.

It is an interesting paper, well written and structured, worth to be published in materials.  However, there are several points that  need to be addressed and revised considerably before acceptance.

Here are few general comments for the authors:
1. It would be nice, if authors would present below.
This is not a real abstract. The authors should construct you abstract in a way to mirror the structure and finds of the paper. This abstract is a shortened description of your results and some kind of explanation of your chosen methodology including experimental data. Your abstract should be a reply to the questions below: context of why your study is important, how you analyzed your data, what are the remarkable finds and how do you interpret these in the light of the background.
2. It would be nice, if authors would show a schematic representation of the topics coverd to improve the impact of this study in a Figure.
3. Introduction must be improved by incorporating more recent references and findings on the topic and by doing a comparison.
4. In conclusion, the disadvantages and limitations of the study should be presented.
5. The English language and grammatical errors should be revised throughout the manuscript.

Author Response

Dear Professor

I am sanding the corrected manuscript

Reviewer 2 Report

Separation efficiency, filtration performance and pressure drop for five filter media specimens were evaluated in this work, in order to select the most suitable filter medium for the inlet air of a motor vehicle’s engine to minimize the engine wear due to dust particles. The manuscript is clearly written and easy to read, with an extensive and detailed research, thus experimental results may be of scientific interest. However, there are some points that need a more detailed explanation.

1) Section 2, pages 7-8. The authors indicate in lines 231-232 that three specimens of each filter element with the same filter medium were tested. Assuming that all filter cartridges were similar, what is the cause of the difference in initial separation efficiency (for a mass loading of dust from 0 to 70 g/m2) that can be seen for the three A filters in Figure 7?

2) Pages 11-12. It would be interesting if the authors could include a figure similar to Figure 10 with the results obtained for filter E, in order to observe the differences (if any) in the number and size of dust particles, similarly to the comparison of particle size distribution for both filters in Figures 11-13.

3) Filter elements with a nanofibre layer and a PTFE layer are more efficient for particle removal but show higher pressure drop increase than the other filters tested, as indicated by the authors in the third conclusion (page 14). Do the authors have any data on the half-life of these new filters compared to that of conventional ones (A and B)?

4) Page 5, Table 1. The permeability data in the first two rows of Table 1 provide the same information, but with different units. Please delete one of those first 2 rows.

5) Reference [25] is not cited in text. Moreover, ref. [59] cited in line 264 (page 9) does not appear in the references section (there are only 57 references).

6) Page 4, line 153. Please, indicate the Qw units (m3/h?).

7) Minor changes:

- Page 6, line 212. Pressure drop is Δpw (like in Eq. (4)), and not Δpf

- Page 6, line 215. The right density units are kg/m3 (or kg m-3), but not kg/m-3

- Page 11, line 347. “Figs. 11 and 12” instead of “Fig. 10 and 11”

- Page 13, Figure 14 caption. “... filter elements A, B, C, D and E.” instead of “... filter elements A, B, C, D.”

Author Response

Dear Proffesor

I am sanding the corrected manuscript
